# Study on the Role of *Cytc* in Response to BmNPV Infection in Silkworm, *Bombyx mori* (Lepidoptera)

**DOI:** 10.3390/ijms20184325

**Published:** 2019-09-04

**Authors:** Xue-Yang Wang, Kang-Hui Wu, Hui-Lin Pang, Ping-Zhen Xu, Mu-Wang Li, Guo-Zheng Zhang

**Affiliations:** 1Jiangsu Key Laboratory of Sericultural Biology and Biotechnology, School of Biotechnology, Jiangsu University of Science and Technology, Zhenjiang 212018, Jiangsu, China (X.-Y.W.) (K.-H.W.) (H.-L.P.) (P.-Z.X.) (M.-W.L.); 2The Key Laboratory of Silkworm and Mulberry Genetic Improvement, Ministry of Agriculture, Sericultural Research Institute, Chinese Academy of Agricultural Science, Zhenjiang 212018, Jiangsu, China

**Keywords:** *Bombyx mori* (*B. mori*), BmNPV, cytochrome *c* (cytc), response mechanism

## Abstract

Bombyx mori nucleopolyhedrovirus (BmNPV) is one of the primary pathogens of the silkworm. *Cytochrome c* (*cytc*) showed a significant response to BmNPV infection in our previous transcriptome study. However, little is known about the role of *Bombyx mori* cytc (Bmcytc) in resistance to BmNPV infection. In this study, the expression levels analysis of *Bmcytc* showed stable expression levels in selected tissues of the resistant strain AN following BmNPV infection, while there was downregulation in the susceptible strain p50, except in the malpighian tubule. To further study the role of *Bmcytc* in viral infection, *Bmcytc* was knocked down with siRNA in vitro, resulting in significant downregulation of selected downstream genes of the mitochondrial pathway, including *Bmapaf*, *Bmcaspase-Nc*, and *Bmcaspase-1*; this was also confirmed by overexpression of *Bmcytc* using the pIZT/V5-His-mCherry insect vector, except *Bmcaspase-1*. Moreover, knockdown of *Bmcytc* significantly promoted the infection process of BmNPV in vitro, while the infection was inhibited by overexpression of *Bmcytc* at the early stage and subsequently increased rapidly. Based on these results, we concluded that *Bmcytc* plays a vital role in BmNPV infection by regulating the mitochondrial apoptosis pathway. Our work provides valuable data for the clarification of the mechanism of silkworm resistance to BmNPV infection.

## 1. Introduction

The silkworm *Bombyx mori* L. (Lepidoptera: Bombycidae) has been domesticated in China for more than 5000 years and generates the main income of some farmers. BmNPV is one of the primary silkworm pathogens that causes serious economic losses every season. Interestingly, certain silkworm strains show high resistance to BmNPV infection [1], but the underlying molecular mechanism is still far from being understood. Moreover, the silkworm is a good model organism for the study of insect genetics and immunology [2,3].

Apoptosis, or programmed cell death (PCD), is a physiological process characteristic of pluricellular organisms. It leads to cellular self-destruction [4,5], which has been reported widely in defense against viral infection [6]. In the mammalian cell, the mitochondrial apoptotic pathway releases several proapoptotic proteins from the intermembrane space of mitochondria [5]. Cytc as a soluble protein is localized in the intermembrane space and is loosely attached to the surface of the inner mitochondrial membrane [7]. Its release from mitochondria is a key event and plays an important role in initiating apoptosis in the mammalian cell [8]. However, the role of cytc is not clear in insect cell apoptosis. It has been reported that cytc is not involved in apoptosis in *Drosophila* by the release from mitochondria [9]. In another study, mitochondria were involved in host defenses against environmental pressure by releasing cytc into the cytoplasm to activate apoptosis in the *Spodoptera litura* and *Spodoptera frugiperda* (Sf9) cell line [10,11]. Therefore, some insects can rely on the mitochondrial apoptosis pathway to protect themselves from an external pressure, such as viral infection.

Cytc is an essential component of the mitochondrial respiratory chain. It has been reported widely that cytc showed a significant response to viral infection. Liu et al., reported that cytc activates the apoptotic signaling pathway to defend against *Anagrapha falcifera* Multiple Nuclear Polyhedrosis Virus (AfMNPV) infection in Lepidopteran *Spodoptera litura* cells [12]. Carthy et al., found that cytc released from mitochondria may be involved in activating caspase after coxsackievirus B3 infection, and this contributes to the loss of host cell viability and progeny virus release [13]. Machida et al., reported that inhibition of cytc release resulted in suppression of Fas-mediated cell death, which could cause hepatitis C virus persistent infection in transgenic mice [14]. In our previous transcriptome study, *cytc* showed a significant response in resistant strain BC9 following BmNPV infection [15], and it might be involved in regulating the mitochondrial apoptosis pathway to defend against BmNPV infection, but that still needs further study. 

According to literature retrieval, there is no related study available to date on *Bmcytc* response to BmNPV infection in the silkworm. In this study, the expression levels of *Bmcytc* in different tissues of different resistant silkworm strains following BmNPV infection were analyzed using RT-qPCR to identify the relationship between *Bmcytc* and BmNPV. To further confirm the role of *Bmcytc* in response to BmNPV infection, the expression of *Bmcytc* was knocked down and overexpressed using siRNA and the insect pIZT/V5-His-mCherry vector in vitro, respectively. Subsequently, the variation of BmNPV infection and the expression levels of selected downstream genes were analyzed after knockdown and overexpression of *Bmcytc* in BmN cells using RT-qPCR. 

## 2. Results

### 2.1. Characterization of the Bmcytc Sequence

The full-length cDNA of *Bmcytc* (GenBank accession number: EU839987) contains a 122 bp 5′-UTR, a 159 bp 3′-UTR, and a complete 327 bp open reading fragment (ORF) which encodes a 108-amino acid protein. The theoretical *p*I and MW are 9.63 and 11.75 kDa, respectively. The functional domain of the Bmcytc protein is the cytochrome_C domain, which is located from amino acids 8 to 107 (Appendix A).

The homologous align analysis showed that the Bmcytc amino acid sequence kept a high conservation among different species (Appendix A). BLASTP blast showed that the Bmcytc amino acid sequence is most similar to that of *Vanessa tameamea* (XP_026495965.1) (96% identity), followed by *Amyelois transitella* (XP_013199433.1), *Danaus plexippus* plexippus (OWR53222.1), *Pieris rapae* (XP_022127978.1) (94.44% identity), *Plutella xylostella* (NP_001292408.1) (93.52% identity), *Hyposmocoma kahamanoa* (XM_026465578.1), *Papilio machaon* (XP_014360069.1) (92.59% identity), *Papilio xuthus* (XP_013179876.1), *Spodoptera litura* (XP_022822995.1) (91.67% identity), *Bactrocera dorsalis* (XP_026321363.1), *Ostrinis furnacalis* (AHV85214.1) (90.74% identity), and *Zeugodacus cucurbitae* (XP_011189550.1) (89.81% identity). Therefore, the high conservation of cytc amino acids among different species shows that Bmcytc may play a vital role in the apoptosis pathway in the silkworm.

The CDS sequence of *Bmcytc* and those in other species were derived from NCBI, which were used to analyze the evolutionary relationships of *cytc* among different species. Based on the DNA/Protein model of Dayhoff + G, a phylogenetic tree including *Bmcytc* and 15 other homologs was generated (Appendix A). *Bmcytc* and these homologous genes were precisely classified into four categories, namely Lepidoptera, Diptera, Rodentia, and Brassicales. The homologous gene sequences of *Bmcytc* in *M. musculus* and *Arabidopsis* shared a low sequence identity with *Bmcytc*, indicating that *cytc* gene in the earlier species might have diverged before the appearance of these orders during evolution.

### 2.2. The Spatio-Temporal Expression Pattern of BMCYTC

To determine the specific biological function of *Bmcytc*, the relative expression levels of *Bmcytc* in different stages and different tissues of the p50 strain were detected by RT-PCR. The highest expression levels of *Bmcytc* were in the head and the lowest were in the hemolymph in different tissues (Figure 1A). In different developmental stages, the highest expression level of *Bmcytc* was in the moth (Figure 1B). During the molting of 4th instar, *Bmcytc* showed a significantly different expression (Figure 1C).

### 2.3. Bmcytc Showed Significant Response to BmNPV Infection in Different Tissues

To determine the response of *Bmcytc* to BmNPV, two different resistant silkworm strains AN and p50 were selected. The resistant levels of selected silkworm strain AN and p50 were determined in our previous study [16,17] the LC_50_ value of AN was >10 ^9^ OB/mL, p50 was about 10 ^5^ OB/mL. The expression patterns of *Bmcytc* in the midgut, hemolymph, malpighian tubule and fat body of different resistant strains following BmNPV infection were analyzed using RT-qPCR (Figure 2). The expression of *Bmcytc* kept a stable level in four selected tissues of the resistant strain AN following BmNPV infection, while it was significantly downregulated in the selected tissues of the susceptible strain p50 following BmNPV infection (except in the malpighian tubule).

### 2.4. Selected Downstream Genes Were Downregulated after Knockdown of BMCYTC in BmN Cells

To further study the molecular mechanism of *Bmcytc* response to BmNPV infection in BmN cells, the siRNA targeting the *Bmcytc* functional domain was used to knock down the expression of *Bmcytc*. The analysis of the effects of siRNA dose indicated that 4 μg of siRNA could be used for RNAi of *Bmcytc* in vitro (Appendix A). The expression levels of *Bmcytc* were determined in BmN cells after transfecting with siRNA at 24 h, 48 h, and 72 h using RT-qPCR. The blank control was transfected with free reagent, the negative control was transfected with siRNA of GFP. The expression of *Bmcytc* was knocked down significantly after transfecting with siRNA at 24 h and reached the lowest at 72 h (Figure 3A).

Referring to the mitochondrial apoptosis pathway of *Drosophila*, three homologous genes of *Drosophila* downstream of *Bmcytc* were selected for further analysis, including *Bmapaf*, *Bmcaspase-Nc*, and *Bmcaspase-1*. BmN cells transfected with siRNA at 24 h, 48 h, and 72 h were collected as a template. The expression level of *Bmapaf* was downregulated at 24 h after knockdown of *Bmcytc* and kept a low expression level at the two selected time points (Figure 3B). The expression levels of *Bmcaspase-1* and *Bmcaspase-Nc* were downregulated at 48 h after knockdown of *Bmcytc* and kept a low expression level at the next two selected time points (Figure 3C,D). 

### 2.5. Knockdown of Bmcytc Promotes BmNPV Infection in BmN Cells

To investigate the role of *Bmcytc* in BmNPV infection, 20 μL of culture medium containing BV-EGFP (1 × 10 ^8^ pfu/mL) was added to BmN cells that had been transfected with siRNA for 24 h. BmNPV infection was recorded using the fluorescence microscope at 24 h, 48 h, and 72 h after inoculation with BV-EGFP while Z-DEVD-FMK was used as positive control. The infection signal appeared early in the siRNA treatment group as compared to the control group at 24 h (Figure 4A), and this was more significant at 48 h and 72 h (Figure 4B,C). To further validate this point, one capsid gene *vp39* of BmNPV was selected to analyze the viral replication in different groups using RT-qPCR. The number of BmNPV in the siRNA treatment group was significantly more than in the control group at 48 h and 72 h (Figure 4D). These data indicated that *Bmcytc* played an important role during BmNPV infection.

### 2.6. Overexpression of Bmcytc Inhibits BmNPV Early Infection in BmN Cells 

To further validate the results of RNAi as described above, *Bmcytc* was overexpressed in BmN cells using the pIZT/V5-His-mCherry vector. The ORF of *Bmcytc* without termination codon was ligated with pIZT/V5-His-mCherry vector between *Kpn* I and *Xba* I. The recombinant overexpression vector was transfected into BmN cells using the transfection reagent, and subsequently 20 μL of culture medium containing BV-EGFP (1 × 10 ^8^ pfu/mL) was added to BmN cells, the pIZT/V5-His-mCherry was used as negative control. BmNPV infection was significantly inhibited after overexpression of *Bmcytc* at 24 h as compared to the control group (Figure 5A,D), while it subsequently increased rapidly at 48 h and 72 h (Figure 5B–D).

### 2.7. Overexpression of Bmcytc Influences the Expression of Selected Downstream Genes in BmN Cells

The expression levels of *Bmcytc* were detected following transfection with pIZT/V5-His-mCherry-Bmcytc in BmN cells using RT-qPCR. The results showed that the expression level of *Bmcytc* was significantly increased in the experimental groups as compared to the control groups at 24 h, 48 h, and 72 h (Figure 6A). The expression levels of *Bmapaf* and *Bmcaspase-Nc* were significantly upregulated after overexpression of *Bmcytc* at 24 h and 48 h, and subsequently back to the basal levels at 72 h (Figure 6B,C). However, the expression of *Bmcaspase-1* showed an opposite trend after overexpression of *Bmcytc* at 48 h and 72 h (Figure 6D).

### 2.8. Analysis of Apoptosis In Vivo and In Vitro

To analyze the relationship between *Bmcytc* and apoptosis during BmNPV infection in vitro, BmN cells containing pIZT/V5-His-mCherry-Bmcytc was incubated with inhibitor Z-DEVD-FMK. The replication of BmNPV was analyzed after overexpression of *Bmcytc* and the treatment with Z-DEVD-FMK using RT-qPCR. The results showed that the number of BmNPV increased after treatment with Z-DEVD-FMK, which indicated that the mitochondrial apoptosis pathway activated by *Bmcytc* could be affected by inhibitor Z-DEVD-FMK (Figure 7A).

To analyze apoptotic responses of cells in NPV infection in vivo, the inhibitor Z-DEVD-FMK and inducer NSC348884 were used to analyze the infection variation of BmNPV in different resistant silkworm strains. The results indicated that the replication of BmNPV was faster in resistant strain AN at 72 h after injection with inhibitor Z-DEVD-FMK than the control group without treatment (Figure 7B). Moreover, the apoptosis induced by inducer NSC348884 could significantly inhibit the replication of BmNPV in susceptible strain p50 at 48 h (Figure 7C). These results indicated that apoptosis plays an important role in response against BmNPV infection.

## 3. Discussion

BmNPV is one of the main pathogens of silkworm disease and causes serious economic losses every year; however, the molecular mechanism of silkworm resistance to BmNPV infection is still unclear. In our previous study, the RNA-Seq transcriptome of the midgut of resistant near-isogenic line BC9 and susceptible recurrent parent p50 following BmNPV infection was selected to systematically screen candidate differentially expressed genes that were involved in resistance to BmNPV infection. Among them, *Bmcytc* showed a significant upregulation in BC9 following BmNPV infection, while it was downregulated in p50, indicating it might play a vital role in BmNPV infection. However, the underlying molecular mechanism of *Bmcytc* in response to BmNPV infection is still unknown. In this study, the analysis of sequence characterization and the role of *Bmcytc* in response to BmNPV infection was studied, and are described as follows.

To analyze the characterization of the nucleotide and amino acid sequence of *Bmcytc*, the biological information methods were adopted. The multiple sequence alignment analysis using DNAMAN 8 software showed that the Bmcytc amino acid sequence shared a high identity with its homologous protein sequence in other species (Appendix A), which was also consistent with the result of phylogenetic tree analysis (Appendix A), indicating its role in the apoptosis pathway. Moreover, the expression profiles of Bmcytc in different developmental stages and tissues were analyzed using RT-qPCR. *Bmcytc* showed a relatively higher expression level during the molting stage, suggesting that the variation of *Bmcytc* during ecdysis might be impacted by molting hormones (Figure 1B,C). Moreover, *Bmcytc* showed higher expression levels in the midgut and head, which might indicate its important role in the process of BmNPV infection in the midgut and head cells (Figure 1A).

Apoptosis is considered to be a means of innate immunity of insects. Early apoptosis has a positive effect on inhibiting viral infection by facilitating the phagocytosis of foreign bodies and causing premature dissociation of infected cells [18]. In this study, the replication of BmNPV was faster in resistant strain AN at 72 h after injection with apoptosis inhibitor, while inhibited in susceptible strain p50 at 48 h after injection with apoptosis inducer (Figure 7B,C), which indicated that apoptosis plays an important role in silkworm larvae in resistance to BmNPV infection. Moreover, the expression levels of *Bmcytc* were stable in the four tissues of the resistant strain AN, while downregulation in the susceptible p50 (except in the malpighian tubule) (Figure 2). These results indicated that the host could rely on the *Bmcytc* active mitochondrial apoptosis pathway to respond to BmNPV infection, which was further validated by overexpression of *Bmcytc* in BmN cells after treatment with inhibitor (Figure 7A).

Our previous transcriptome data showed that *Bmcytc* and its downstream key genes were related to BmNPV infection [15]. Combining with relevant reports, we presumed that *Bmcytc* is involved in response to BmNPV infection by activating the mitochondrial apoptosis pathway to promote premature dissociation of infected BmN cells. To prove the hypothesis, the specific siRNA was used to knock down the expression of *Bmcytc* in BmN cells. The results showed that knockdown of *Bmcytc* not only downregulated the expression of its downstream key genes, including *Bmapaf*, *Bmcaspase-Nc* and *Bmcaspase-1* (Figure 3), but also promoted BmNPV infection in BmN cells (Figure 4), indicating that our inference is reasonable. 

Furthermore, to further confirm the variation of BmNPV infection and the expression of downstream key genes following RNAi of *Bmcytc* in vitro as described above, *Bmcytc* was overexpressed using the pIZT/V5-His-mCherry insect vector by inserting the ORF of *Bmcytc* and transfecting in BmN cells (Figure 5A–C). The expression level of *Bmcytc* was upregulated for 3.5 times at 24 h following transfection with pIZT/V5-His-mCherry-Bmcytc as compared with control, which was still increased at 48 h and 72 h (Figure 6A). Moreover, the expression levels of its downstream key genes were upregulated in the transgenic cell lines (Figure 6B–D), except *Bmcaspase-1*, possibly because it did not play an important role in the mitochondrial apoptosis pathway as reported in other species. The defense of the silkworm against pathogens is through the immune system, including apoptosis and active substances, such as antibacterial peptides [19]. However, in the later stages of infection, apoptosis serves as part of the viral strategy to self-propagate and promotes dissemination within the host [20]. This might explain the significant inhibition of BmNPV infection in BmN cells that were transfected with overexpression vector at 24 h, subsequently increasing rapidly at 48 h and 72 h (Figure 5D). According to the study as described above, we reached an agreement that *Bmcytc* plays a crucial role in the response against BmNPV infection by activating the mitochondrial apoptosis pathway. Taken together, the role of *Bmcytc* in response to BmNPV infection will make an important contribution in elucidating the molecular mechanism of silkworm resistance.

## 4. Materials and Methods

### 4.1. Silkworm and BmNPV

The susceptible strain p50 and the resistant strain AN were maintained in the Key Laboratory of Sericulture, School of Life Sciences, Jiangsu University of Science and Technology University, Zhenjiang, China. The first three instar larvae were reared on a fresh artificial diet at 26 ± 1 °C, 75 ± 5% relative humidity, and a 12 h day/night cycle. The rearing temperature for the last two instars was reduced to 24 ± 1 °C, but the other conditions remained unchanged.

All larvae were starved for 24 h at the first day of the fifth instar. Thirty larvae from each group were fed with 5 μL of BmNPV suspended in sterile water (1.0 × 10 ^5^ OB/mL) per larva, sterile water was used as control. Budded virus containing EGFP-tagged (BV-EGFP) BmNPV were maintained in our laboratory. Polyhedrin promoter was used, the EGFP was inserted between *BamH* I and *Xho* I, which is not fused with any protein. The amount of BV-EGFP (pfu/mL) was determined as described in our previous study [21]. The equal volume culture containing BV-EGFP (1 × 10 ^8^ pfu/mL) was added into the medium to conduct the infection process in different treatment groups.

### 4.2. Bioinformatics Analysis

The DNAMAN 8.0 software (Lynnon Corporation, Quebec, Canada) was used to analyze the cDNA and deduce the protein sequence of Bmcytc. The SMART server (http://smart.embl-heidelberg.de/) was used to predict its conserved motif. The BLASTP tool (http://www.ncbi.nlm.nih.gov/) was used to analyze the sequences of orthologs. The MUSCLE module of the MEGA7 software was used to align the amino acid sequences among different species. A maximum-likelihood tree was generated using MEGA7 with a bootstrap of 1000 replications, and the best DNA/Protein model of Dayhoff + G was adopted.

### 4.3. Sample Preparation, RNA Extraction and cDNA Synthesis

Silkworms at the first day of 5th instar were starved for 24 h and then fed with 5 μL BmNPV (1.0 × 10^5^ OB/mL) per larva. The samples were collected after BmNPV infection at 24 h. The head, midgut, hemolymph, fat body, malpighian tubule and epidermis of 30 larvae were dissected and mixed together to minimize individual genetic differences. The whole bodies of thirty larvae in different developmental stages were collected. All samples were frozen immediately in liquid nitrogen and stored at −80 °C.

The total RNA of the silkworm tissues was extracted with TRIzol Reagent (Invitrogen, New York, USA) according to the manufacturer’s instructions. A NanoDrop 2000 spectrophotometer (Thermo Fisher Scientific, New York, NY, USA) was used to quantify the RNA concentrations and purity. The RNA integrity was confirmed by 1% agarose gel electrophoresis. The first strand cDNA was synthesized using an RT reagent kit (TaKaRa Biotechnology Co. Ltd., Dalian, China) according to the manufacturer’s instructions.

### 4.4. Quantitative Reverse Transcription PCR (RT-qPCR)

The expression level of *Bmcytc* was determined by RT-qPCR. All primer sequences are listed in Table 1. RT-qPCR reactions were prepared with the NovoStart^®^SYBR qPCR SuperMix Plus (Novoprotein, Nanjing, China) by following the manufacturer’s instructions. Reactions were carried out using the LightCycler^®^ 96 System (Roche, Switzerland, France). The thermal cycling profile consisted of an initial denaturation at 95 °C for 5 min, then 40 cycles at 95 °C for 5 s and 60 °C for 31 s. All samples were performed in triplicate. Relative expression levels were calculated using the 2^−ΔΔCT^ method according to the protocol described by Livak et al. [22]. *B. mori* glyceraldehyde-3-phosphate dehydrogenase (*BmGAPDH*) was used as an internal control [23]. Statistical analysis was performed using GraphPad Prism 5 software (GraphPad Software, San Diego, USA). The *t*-test method was used to analyze the acquired data. A *p*-value of <0.05 was considered to be statistically significant.

### 4.5. Synthesis of siRNA

To knockdown the expression of *Bmcytc* in BmN cells, two specific targets in the functional domain of *Bmcytc* were selected. The specific target of green fluorescent protein (GFP) was used as negative control. The siRNA oligos were designed and synthesized by Sangon Biotechnology (Table 2). The In Vitro Transcription T7 Kit (for siRNA synthesis) (TaKaRa Biotechnology Co. Ltd., Dalian, China) was used to synthesize siRNA according to the manufacturer’s instructions. The concentration and purity of the siRNA were measured by a NanoDrop 2000 spectrophotometer (Thermo Fisher Scientific, New York, NY, USA). The integrity of the siRNA was determined by 3% agarose gel electrophoresis. The siRNA with good quality was stored at −80 °C until use.

### 4.6. Construction of pIZT/V5-His-mCherry-Bmcytc Overexpression Vector 

The ORF of *Bmcytc* without a termination codon was amplified from p50 midgut cDNA with the primers *Bmcytc KX* (Table 1; the underlined portions indicate the *Kpn* I and *Xba* I restriction sites, respectively). The purified PCR products were ligated with pMD-19T vector for sequencing. The ORF of *Bmcytc* and the pIZT/V5-His-Mcherry vector were digested with *Kpn* I and *Xba* I (TaKaRa Biotechnology Co. Ltd., Dalian, China) and then ligated with T4 DNA ligase (TaKaRa Biotechnology Co. Ltd., Dalian, China). The recombinant expression vector pIZT/V5-His-mCherry-Bmcytc was confirmed by *Kpn* I and *Xba* I enzyme digestion and sequencing by Sangon Biotechnology. 

### 4.7. BmN Cell Culture and Transfection 

The silkworm ovarian cell line, BmN, was cultured in TC-100 (AppliChem, Germany) supplemented with 10% (*v*/*v*) fetal bovine serum (FBS) (Thermo Scientific, New York, USA), 200 µg/mL penicillin, and 100 µg/mL streptomycin at 28 °C. siRNA and overexpression vector were transfected with Neofect^TM^ DNA transfection reagent (NEOFECT, Beijing, China) by following the manufacturer’s instructions. Briefly, BmN cells were seeded into the 3 cm^2^ culture flasks (approximately 1 × 10 ^5^ cells/well) before transfection. 4 µg siRNA or vector and 4 µL transfection reagent were added successively into 200 µL serum-free TC-100 to prepare transfection solution, which was added subsequently into the culture medium. 

Cellular fluorescence images were taken using a Leica inverted research grade microscope DMi3000B camera and processed with the Leica Application Suite V4.6 software (Leica, Wetzlar, Germany).

### 4.8. Inhibition and Induction of Apoptosis

Z-DEVD-FMK and NSC348884 reagents (Topscience, Beijing, China) were used for inhibition and induction of apoptosis, respectively. The final concentrations 5 µM and 10 µM of NSC348884 and Z-DEVD-FMK were selected according to the manufacturers’ instruction, respectively. The inhibition and induction effects were analyzed at 24 h, 48 h, and 72 h after treatment with Z-DEVD-FMK and NSC348884. 

### 4.9. Genomic DNA Extraction

Genomic DNA was extracted from the whole body of silkworm larvae and BmN cells with DNA extraction buffer (containing 100 mM Tris-HCl pH 7.5, 100 mM ethylenediaminetetra-acetic acid, 100 mM NaCl, and 0.5% sodium dodecyl sulfate), incubated in 65 °C for 30 min, mixed with the mixture of lithium chloride monohydrate (LiCl)/potassium Acetate (KAc) (containing 4.3 M LiCl and 1.4 M KAc), purified with a isopropanol and 75% ethyl alcohol precipitation extraction, followed by RNaseA treatment. The integrity, purity, and concentration of DNA were generated as description of RNA above. 

## Figures and Tables

**Figure 1 ijms-20-04325-f001:**
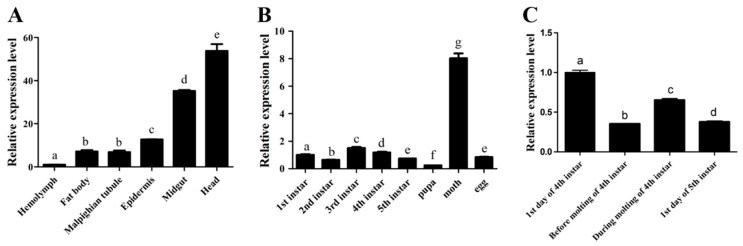
The spatiotemporal expression analysis of *Bmcytc*. (**A**) The expression levels of *Bmcytc* in different tissues. (**B**) The expression levels of *Bmcytc* in different developmental stages. (**C**) The expression levels of *Bmcytc* during molting stages. The data were normalized using *BmGAPDH* and are represented as the mean ± standard error of the mean, from three independent experiments. Relative expression levels were calculated using the 2^−ΔΔCt^ method. Statistical analysis was performed using GraphPad Prism 5 software. The difference between the samples was analyzed by the *t*-test method. Significant differences are indicated by different letters, e.g., a, b, c (*p* <0.05).

**Figure 2 ijms-20-04325-f002:**
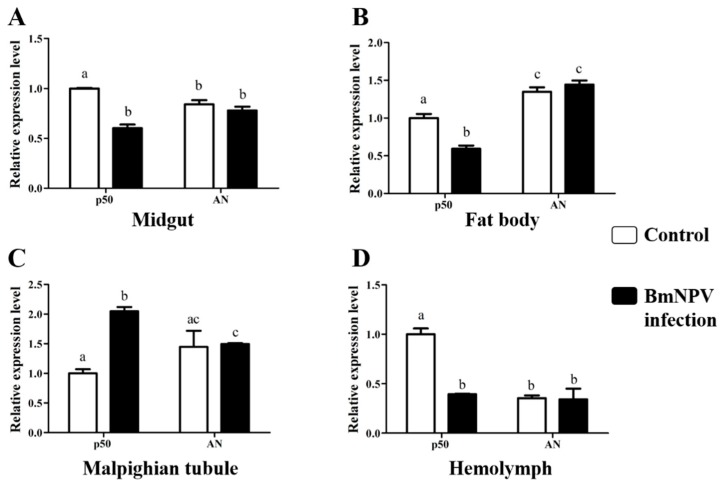
Expression analysis of *Bmcytc* in different tissues of different resistant strains following BmNPV infection. Expression levels of *Bmcytc* in midgut (**A**), fat body (**B**), malpighian tubule (**C**), and hemolymph (**D**) at 24 h following BmNPV infection. The data were normalized using *BmGAPDH* and are represented as the mean ± standard error of the mean, from three independent experiments. Relative expression levels were calculated using the 2^−ΔΔCt^ method. Statistical analysis was performed using GraphPad Prism 5 software. The difference between the samples was analyzed by the *t*-test method. Significant differences are indicated by different letters, e.g., a, b, c (*p* < 0.05).

**Figure 3 ijms-20-04325-f003:**
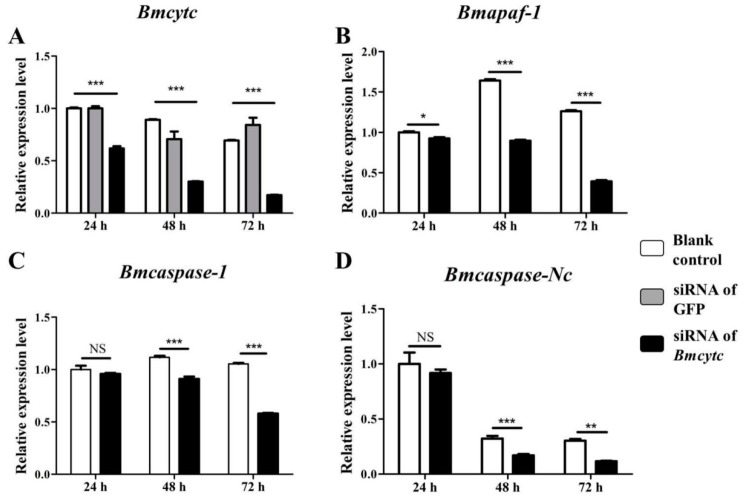
Expression analysis of selected downstream genes after knockdown of *Bmcytc* at different times. (**A**) The expression analysis of *Bmcytc* after transfecting with siRNA at different times, siRNA of GFP was used as negative control. The expression analysis of *Bmapaf* (**B**), *Bmcaspase-NC* (**C**), and *Bmcaspase-1* (**D**) after knockdown of *Bmcytc* at different times. The data were normalized using *BmGAPDH* and are represented as the mean ± standard error of the mean, from three independent experiments. Relative expression levels were calculated using the 2^−ΔΔ*C*t^ method. Statistical analysis was performed using GraphPad Prism 5 software. The difference between the samples was analyzed by the *t*-test method. Significant differences are indicated by asterisks (*p* <0.05). NS, not significant. *, *p* < 0.05; **, *p* < 0.01; ***, *p* < 0.001.

**Figure 4 ijms-20-04325-f004:**
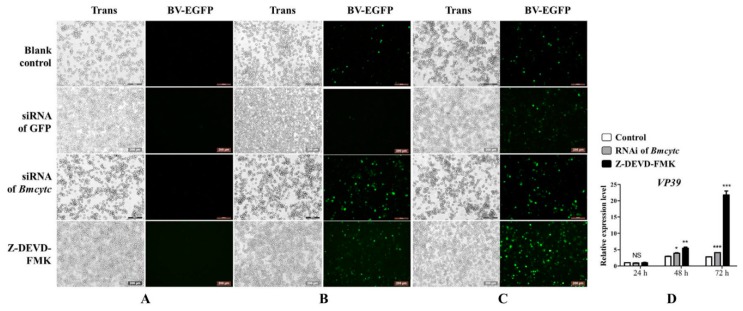
Analysis of BmNPV infection in BmN cells following knockdown of *Bmcytc* at different times. (**A**) 24 h after BV-EGFP infection. (**B**) 48 h after BV-EGFP infection. (**C**) 72 h after BV-EGFP infection. (**D**) The expression analysis of *vp39* after knockdown of *Bmcytc* at different times. Scale bar = 200 μm. Trans (white), optical transmission. EGFP (Green), expressed following the replication of BV. siRNA of GFP was used as negative control, Z-DEVD-FMK was used as positive control. The data were normalized using *BmGAPDH* and are represented as the mean ± standard error of the mean, from three independent experiments. Relative expression levels were calculated using the 2^−ΔΔCt^ method. Statistical analysis was performed using GraphPad Prism 5 software. The difference between the samples was analyzed by the *t-*test method. Significant differences are indicated by asterisks (*p* < 0.05). NS, not significant. *, *p* < 0.05; **, *p* < 0.01; ***, *p* < 0.001.

**Figure 5 ijms-20-04325-f005:**
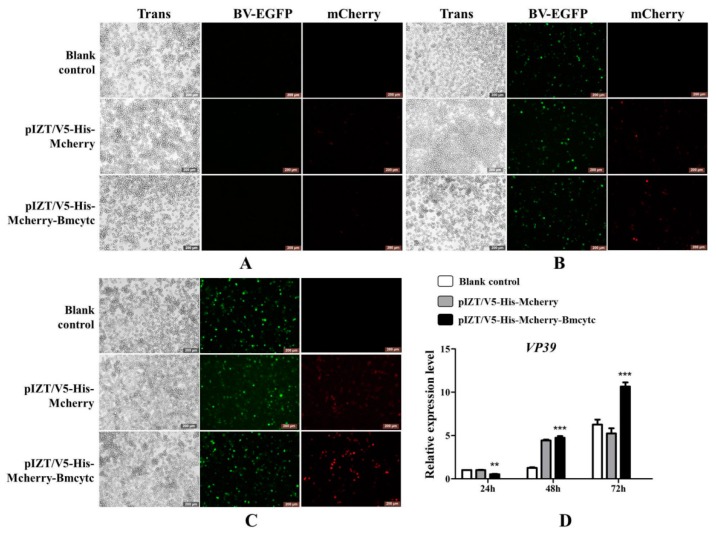
Analysis of BmNPV infection after overexpression of *Bmcytc* in BmN cells at different times. (**A**) 24 h after BV-EGFP infection. (**B**) 48 h after BV-EGFP infection. (**C**) 72 h after BV-EGFP infection. (**D**) The expression analysis of *vp39* after knockdown of *Bmcytc* at different times. Scale bar = 200 μm. Trans (white), optical transmission. EGFP (Green), expressed following the replication of BV. mCherry (Red), fused expression with Bmcytc protein. pIZT/V5-His-mCherry was negative control. The data were normalized using *BmGAPDH* and are represented as the mean ± standard error of the mean, from three independent experiments. Relative expression levels were calculated using the 2^−ΔΔCt^ method. Statistical analysis was performed using GraphPad Prism 5 software. The difference between the samples was analyzed by the *t*-test method. Significant differences are indicated by asterisks (*p* <0.05). NS, no significant. *, *p* < 0.05; **, *p* < 0.01; ***, *p* < 0.001.

**Figure 6 ijms-20-04325-f006:**
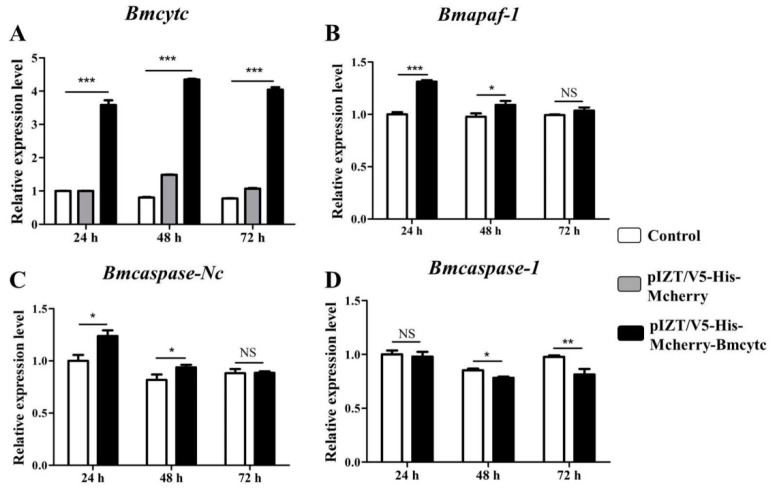
Expression analysis of selected downstream genes after overexpression of *Bmcytc* at different times. (**A**) The expression levels of *Bmcytc* after transfecting with overexpression vector at different times, pIZT/V5-His-mCherry was used as negative control. The expression analysis of *Bmapaf* (**B**) *Bmcaspase-NC* (**C**) and *Bmcaspase-1* (**D**) after overexpression of *Bmcytc* at different times. The data were normalized using *BmGAPDH* and are represented as the mean ± standard error of the mean, from three independent experiments. Relative expression levels were calculated using the 2^−ΔΔCt^ method. Statistical analysis was performed using GraphPad Prism 5 software. The difference between the samples was analyzed by the *t*-test method. Significant differences are indicated by asterisks (*p* < 0.05). NS, not significant. *, *p* < 0.05; **, *p* < 0.01; ***, *p* < 0.001.

**Figure 7 ijms-20-04325-f007:**
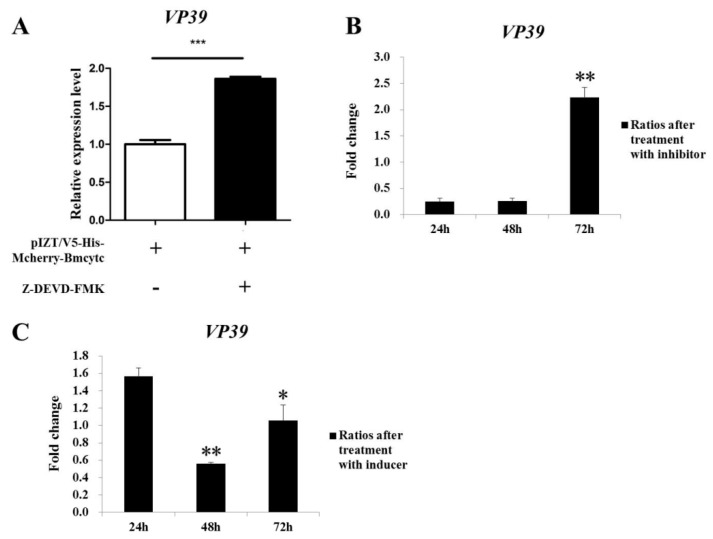
Analysis of the replication of BmNPV in vivo and in vitro. (**A**) The replication of BmNPV in BmN cells after overexpression of *Bmcytc* and treatment with inhibitor Z-DEVD-FMK. The replication of BmNPV in different silkworm strains after treatment with inhibitor (**B**) and inducer (**C**) at different times. The ratios obtained by treatment with inhibitor or inducer against injected with sterile water. The data were normalized using *BmGAPDH* and are represented as the mean ± standard error of the mean, from three independent experiments. Relative expression levels were calculated using the 2^−ΔΔCt^ method. Statistical analysis was performed using GraphPad Prism 5 software. The difference between the samples was analyzed by the *t*-test method. Significant differences are indicated by asterisks (*p* < 0.05). NS, not significant. *, *p* < 0.05; **, *p* < 0.01; ***, *p* < 0.001.

**Table 1 ijms-20-04325-t001:** List of primers used in this study.

Gene Name	Forward Primers (5′-3’)	Revers Primers (5′-3’)
*Bmcytc*	TCATACTCCGATGCCAATAAAGC	CATTTGCCTTCTTGAGTCCAGC
*BmApaf*	TCACAACCCTCTAAAATCACACCAG	CGACAGCCAGTAATGGGTGTATGAG
*BmCaspase-Nc*	GAGGACGATGTGAGCAGGGAT	TTCAGCAGGAACGAAATGTAGC
*BmCaspase-1*	AACGGCAACGAAGACGAAGG	GGTGCCCGTGCGAGATTTTA
*BmGAPDH*	CGATTCAACATTCCAGAGCA	GAACACCATAGCAAGCACGAC
*VP39*	CAACTTTTTGCGAAACGACTT	GGCTACACCTCCACTTGCTT
*Bmcytc KX*	GGGGTACCATGGGTGTACCTGCAGGAAA	GCTCTAGACTTGGTAGCAGATTTGAGATAGG

**Table 2 ijms-20-04325-t002:** Primers used to synthesize siRNA.

Primer Names	Sequences (5′-3′)
Bmcyto-1 Olig-1	GATCACTAATACGACTCACTATAGGGGGACCGAATCTACATGGATTT
Bmcyto-1 Olig-2	AAATCCATGTAGATTCGGTCCCCCTATAGTGAGTCGTATTAGTGATC
Bmcyto-1 Olig-3	AAGGACCGAATCTACATGGATCCCTATAGTGAGTCGTATTAGTGATC
Bmcyto-1 Olig-4	GATCACTAATACGACTCACTATAGGGATCCATGTAGATTCGGTCCTT
Bmcyto-2 Olig-1	GATCACTAATACGACTCACTATAGGGCCTTATTGCCTATCTCAAATT
Bmcyto-2 Olig-2	AATTTGAGATAGGCAATAAGGCCCTATAGTGAGTCGTATTAGTGATC
Bmcyto-2 Olig-3	AACCTTATTGCCTATCTCAAACCCTATAGTGAGTCGTATTAGTGATC
Bmcyto-2 Olig-4	GATCACTAATACGACTCACTATAGGGTTTGAGATAGGCAATAAGGTT
GFP Olig-1	GATCACTAATACGACTCACTATAGGGGGAGTTGTCCCAATTCTTGTT
GFP Olig-2	AACAAGAATTGGGACAACTCCCCCTATAGTGAGTCGTATTAGTGATC
GFP Olig-3	AAGGAGTTGTCCCAATTCTTGCCCTATAGTGAGTCGTATTAGTGATC
GFP Olig-4	GATCACTAATACGACTCACTATAGGGCAAGAATTGGGACAACTCCTT

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
