# Peer review of "Study on the Role of Cytc in Response to BmNPV Infection in Silkworm, Bombyx mori (Lepidoptera)"

_ijms, 2019, doi:10.3390/ijms20184325_

Round 1

Reviewer 1 Report

This manuscript presents data on a role of Bombyx mori cytc(Bmcytc) in resistance to Bombyx mori nucleopolyhedrovirus infection. Authors compared expression levels of Bmcytc and apoptosis-related genes of NPV-resistant and susceptible B. mori strains. In addition, knockdown and over-expression of Bmcytc were carried out in vitro, BmN cells. They concluded that Bmcytc plays a role in BmNPV infection by regulating mitochondrial apoptosis pathway.

Major comments:

Conclusion from Bmcytc knockdown and overexpression experiments is not convincing. The authors do not wipe out suspicion that phenotypes observed in the experiment are due to off-target effects or some kind of effect by siRNA transfection itself. Additional experiments should be needed: I recommend use of appropriate negative control (siRNA not specific to the target), conducting independent experiments using different target site sequences and collecting data on dose-response (effects of siRNA dose). Similarly, overexpression experiment should have been carried out with an appropriate negative control.

To make conclusion of this manuscript reasonable, apoptotic responses of cells in NPV infection have to be analyzed in vitro (control vs knockdown, control vs overexpression) and in vivo (NPV-resistant vs susceptible strain).

Other comments

I think readers would want to know why authors used strain AN not BC9. Please write the reason(s). Additionally, please kindly provide more detailed information on resistance of AN strain to BmNPV in the text.

Please follow the ICTV's recommendation for how to write a virus name (https://talk.ictvonline.org/files/ictv_documents/m/gen_info/7004). For example, Bombyx mori nucleopolyhedrovirus in abstract should be corrected. Please check throughout text about virus names.

I’d like to make sure that phylogenetic inference was performed using NJ method. How did authors use the best model, LG+G, in NJ?

Which version of MEGA was used, 6 or 7? (MEGA6 in the text, but MEGA7 in supplementary materials.)

6. L56, actives; activates?

Author Response

Thank you for your professional suggestion, we have revised our manuscript after having carefully considered the comments, details are as follows:

Major comments:

Conclusion from Bmcytc knockdown and overexpression experiments is not convincing. The authors do not wipe out suspicion that phenotypes observed in the experiment are due to off-target effects or some kind of effect by siRNA transfection itself. Additional experiments should be needed: I recommend use of appropriate negative control (siRNA not specific to the target), conducting independent experiments using different target site sequences and collecting data on dose-response (effects of siRNA dose). Similarly, overexpression experiment should have been carried out with an appropriate negative control.

Reply:

1) The siRNA targeted to GFP has been selected as negative control, the parallel experiments have been supplemented at 24 hr, 48 hr, 72 hr. The results showed that there were no phenotypes due to off-target effects or some kind of effects by siRNA transfection itself, which also have been added into manuscript (Line 145).

2) The effects of siRNA dose have been supplemented. The different concentrations of siRNA including 2 μg, 4 μg, and 6 μg were selected for further validation . The results showed that the siRNA of Bmcytc did not show any dose effects at 24 hr after transfection (Fig. S4), it is reasonable to use 4 μg of siRNA in this study, which is also referenced to manufacturer’s instruction. The description has been added into manuscript (Line 142). The detail data have been uploaded in the Supplementary Files.

3) The appropriate negative control used the pIZT/V5-His-mCherry vector have been supplemented at 24 hr, 48 hr, 72 hr. The results that there were no phenotypes due to some kind of effect by pIZT/V5-His-mCherry vector transfection itself, which also have been added into manuscript (Line 193).

To make conclusion of this manuscript reasonable, apoptotic responses of cells in NPV infection have to be analyzed in vitro (control vs knockdown, control vs overexpression) and in vivo (NPV-resistant vs susceptible strain).

Reply:

1) To analyze the apoptotic responses of cells in NPV in vitro, the replication of BmNPV was analyzed after treatment with inhibitor in the BmN cells that contained pIZT/V5-His-mCherry-Bmcytc. The results showed that the apoptotic responses activated by Bmcytc were involved in BmNPV infection, which has been added into manuscript (Line 226-231).

2) To determine the apoptotic responses of cells in NPV infection in vivo, the inhibitor Z-DEVD-FMK and inducer NSC348884 mixed with BV were injected into silkworm larvae, the virus proliferation was analyzed using RT-qPCR. The replication of BmNPV was faster in resistant strain AN at 72 hr after injected with inhibitor Z-DEVD-FMK as compared with control group without treatment, which has been added into manuscript (Line 232-237).

Other comments:

I think readers would want to know why authors used strain AN not BC9. Please write the reason(s). Additionally, please kindly provide more detailed information on resistance of AN strain to BmNPV in the text.

Reply: The BC9 strain was used in first author’s PhD study, which was not allowed to use after graduation. Bmcytc was identified by comparing the transcriptome data of p50 (susceptible strain) and BC9 (resistant strain) following BmNPV infection. To further study the role of Bmcytc in response to BmNPV infection, a resistant strain AN was selected. The description of resistant levels of the two strains was added into manuscript (Line 121).

Please follow the ICTV's recommendation for how to write a virus name (https://talk.ictvonline.org/files/ictv_documents/m/gen_info/7004). For example, Bombyx mori nucleopolyhedrovirus in abstract should be corrected. Please check throughout text about virus names.

Reply: The virus name used in this study has been revised following the ICTV's recommendation.

I’d like to make sure that phylogenetic inference was performed using NJ method. How did authors use the best model, LG+G, in NJ?

Reply: Yes, we used NJ method to analyze the phylogenetic tree. Before construction of the phylogenetic, we used the “DNA/Protein model” in MEGA 7 software to select the best model (LG+G) that suit for generating the phylogenetic tree.

Which version of MEGA was used, 6 or 7? (MEGA6 in the text, but MEGA7 in supplementary materials.)

Reply: The version of MEGA7 was used in this study, which have been revised in the full text.

L56, actives; activates?

Reply: It has been replaced with activates.

Reviewer 2 Report

This is a very interesting study on the role of Bmcytc on BmNPV infection. The paper is well written, although the discussion is rather limited, as well as the bibliography. In addition, the results are heavily relying on the evaluation of gene expression by RT-PCR. The authors should verify their key results with alternative ways of measuring apoptosis during BmNPV infection as Bmcytc may have alternative roles. In addition, authors should also use apoptosis inhibitors to verify that the effect of knocking down Bmcytc on BmNPV infection is indeed mediated bythe role of Bmcytc on apoptosis.

Author Response

Thank you for your professional suggestion, we have revised our manuscript after having carefully considered the comments, details are as follows:

Major concern:

This is a very interesting study on the role of Bmcytc on BmNPV infection. The paper is well written, although the discussion is rather limited, as well as the bibliography. In addition, the results are heavily relying on the evaluation of gene expression by RT-PCR. The authors should verify their key results with alternative ways of measuring apoptosis during BmNPV infection as Bmcytc may have alternative roles. In addition, authors should also use apoptosis inhibitors to verify that the effect of knocking down Bmcytc on BmNPV infection is indeed mediated by the role of Bmcytc on apoptosis.

Reply:

1) To further validate the role of Bmcytc in response to BmNPV infection. The replication of BmNPV was analyzed after treatment with inhibitor in the BmN cells that contained pIZT/V5-His-mCherry-Bmcytc. The results showed that the apoptotic responses activated by Bmcytc were involved in BmNPV infection, which has been added into manuscript. Moreover, the inhibitor Z-DEVD-FMK and inducer NSC348884 mixed with BV were injected into silkworm larvae. The results showed that the replication of BmNPV was faster in resistant strain AN at 72 hr after injected with inhibitor Z-DEVD-FMK as compared with control group without treatment. These results have been added into manuscript to support our conclusion (Line 226-237).

2) The apoptosis inhibitor used as positive control has been supplemented. The results showed that the variations of BmNPV infection and the downstream key genes causing by apoptosis inhibitor was basically consistent with knockdown of Bmcytc, which have been added into the manuscript (Line 145).

Round 2

Reviewer 1 Report

This version of manuscript has been improved over previous one. But, let me please ask again about the procedure for phylogenetic tree construction. In my knowledge, MEGA7 doesn’t provide LG+G as a substitution model option in NJ method. Please confirm the procedure used. I suspect other method, for example, ML, may be operated.

Author Response

Thank you for your professional suggestion, the phylogenetic tree has been generated again using maximum-likelihood method. The description has been revised in the manuscript.

Reviewer 2 Report

The error bars in Figures 7C and 7D are missing.

Author Response

Thank you for your professional suggestion, the error bars have been added into the Figures 7.